# Spatiotemporal Distribution of Irrigation Water Use Efficiency from the Perspective of Water Footprints in Heilongjiang Province

**Wei Liu** [1,2,*], **Ziao Ma** [3] **and Bo Lei** [1,2]

1 Department of Irrigation and Drainage, China Institute of Water Resources and Hydropower Research, Beijing 100048, China; leibo@iwhr.com
2 State Key Laboratory of Simulation and Regulation of Water Cycle in River Basin, China Institute of Water Resources and Hydropower Research, Beijing 100038, China
3 Construction and Administration Bureau of South-to-North Water Diversion Middle Route Project Beijing, Beijing 100038, China; mza1856987@126.com
* Correspondence: liuwei@iwhr.com

**Abstract:** Water footprints can reflect the sources and utilities of water resources. Introducing the water footprint theory to evaluate irrigation water use efficiency can reflect agricultural water consumption more scientifically and accurately. This study analyzes the variation trends of the blue, green and gray water footprints of grains in different regions of Heilongjiang Province and selects the grain-sowing area, total agricultural machinery power, grain blue water footprint and green water footprints and absolute fertilizer amount as input indexes and the agricultural gross product and gray water footprint of grain as output indexes. A slacks-based measure–data envelopment analysis (SBM-DEA) model is used to estimate the irrigation water use efficiencies of 11 cities in Heilongjiang Province, analyze the corresponding spatiotemporal distribution and further decompose and calculate the irrigation water use efficiencies of the five economically underdeveloped second-level cities. The results suggest that the spatial distribution of the grain water footprint in Heilongjiang Province reflects coexisting areas of excess and scarcity. The irrigation water use efficiency showed a steady and slow downward trend from 2008 to 2018. The irrigation water use efficiency reflected significant spatial differences in Heilongjiang Province, with a pattern of high values in the southwest and low values in the northeast; these differences have gradually narrowed. The average irrigation water use efficiency in Heilongjiang Province was 0.821 and the irrigation water efficiencies of Harbin, Qiqihar and Jixi were at the forefront of the province. Jiamusi, Hegang, Shuangyashan, Yichun, and Mudanjiang are the five cities with below-provincial-average irrigation water use efficiencies. The irrigation water use efficiency of Heilongjiang Province mainly depends on the pure technical efficiency. In the future, technical inputs should be improved on the basis of optimizing the agricultural production layout, focusing on improving the pure technical efficiency. The research results obtained herein can provide a theoretical basis for agricultural water management in Heilongjiang Province.

**Keywords:** Heilongjiang Province; water footprint; SBM-DEA model; irrigation water efficiency; spatiotemporal distribution

## 1. Introduction

Agriculture is the basic industry of the national economy, and the contradiction between agricultural water demands (the largest water sector in China) and industrial and domestic water use directly affects the sustainable development of water resources and the steady development of the agricultural economy in China [1]. Heilongjiang Province is the largest commodity grain base in China, and the total grain output by this province had achieved 16 consecutive years of growth by 2020. The irrigation area in Heilongjiang

Province is $3.875 \times 10^6$ hm$^2$. The main irrigation method is deficit irrigation, and the water-saving technologies include control irrigation technology, pipeline water-saving measures, etc. To positively respond to national food security policy and steadily increase the grain output, problems associated with the grain output growth and agricultural water supply shortages in Heilongjiang Province urgently need to be solved. Heilongjiang Province is experiencing water shortage with uneven temporal and spatial distribution; in addition, the irrigation water use efficiency in some areas is low due to the aging irrigation facilities and weak awareness of water saving. These have gradually become the main factors restricting the agricultural development of Heilongjiang Province. It is urgent to improve the irrigation water use efficiency and effectively relieve the pressures associated with irrigation water demands [2,3].

At present, irrigation water use efficiency has been analyzed from many research angles and methods that can be summarized in the following two aspects: (1) Evaluations of irrigation water use efficiency according to experimental observations; for example, Shao Dongguo et al. [4] quantitatively described the regional water conversion relationship by observing a water balance test in a typical area of 1000 ha and introduced an "irrigation water intake of 1000 ha" as a new index for evaluating irrigation water use efficiency. Cao Xinchun et al. [5] constructed an index system to evaluate field irrigation water use efficiency based on the water footprint theory; thus providing a new perspective for evaluating irrigation water use efficiency. (2) The second aspect involves calculating irrigation water use efficiency using input–output methods, including data envelopment analysis (DEA) and stochastic frontier analysis (SFA) methods [6,7]. DEA methods are more practical and have become the most commonly used method to evaluate irrigation water use efficiency because these methods avoid the influence of subjective factors and do not require any predetermined function [8]. For example, Sun Fuhua et al. constructed an evaluation model based on DEA-Malmquist to calculate the irrigation water use efficiency of 31 provinces in China from 2011 to 2015 [9]. KH Choi et al. investigated changes in the productivity of Jeollabuk-do local water supply service by using analysis of efficiency by DEA and MIA [10]. Ja Palomero Gonzalez et al. provided a methodology to evaluate the efficiency of the sectors of a water distribution network by applying a data envelopment analysis weighted Russell directional distance (DEA-WRDD) model [11].

The main goal of most previous studies was to obtain the irrigation water use efficiency value without analyzing the corresponding spatial distribution, so it is difficult to propose applicable water-saving schemes for different regions using existing research results. Irrigation water use efficiency is usually evaluated only with input factors such as labor, funds and water resources, while the unexpected output is not usually considered; thus, the results do not conform to the actual situation.

The water footprint theory provides a new idea for the evaluation and management of water resource use efficiency. Some scholars have carried out research in which irrigation water use efficiency was evaluated based on the water footprint theory. However, the general related research has considered only blue water and green water, while gray water has not been included, causing the calculation results to deviate from the actual situation [12]. Aiming to solve the problems described above, this research constructs a comprehensive evaluation index system encompassing the grain-sowing area, total agricultural machinery power, grain blue water footprint, grain green water footprint and absolute chemical fertilizer amount as the input indexes; the agricultural gross product and grain gray water footprint are identified as output indexes. The data envelopment analysis slacks-based model (DEA-SBM) is used to calculate the irrigation water efficiency 11 cities of Heilongjiang Province in 2008~2018, and the spatial distribution presented and analyzed. The results of this research have important practical significance for improving irrigation water use efficiency and ensuring food and water security.

## 2. Materials and Methods

### 2.1. Overview of the Study Area

Heilongjiang Province is located in Northeast China (121°11′–135°05′ E, 43°26′–53°33′ N) and is the largest commodity grain production base in China. Maize, soybean and rice are main crops. The yields of the three crops were 49.97%, 9.3% and 38.05%, respectively, in 2017 [13]. Irrigation water is the main agricultural water resource consumed in Heilongjiang Province, accounting for 89.63% of the total agricultural water consumption. The total amount of water resources is not high in this region, and the spatial and temporal differences manifested in the water resource distribution are substantial [14,15]. The research object contains 11 cities in Heilongjiang Province: Harbin, Qiqihar, Daqing, Yichun, Jiamusi, Jixi, Mudanjiang, Hegang, Shuangyashan, Qitaihe and Suihua (Figure 1). Daxinganling and Heihe are not part of the research object in Heilongjiang Province due to their terrain, climate conditions and other factors.

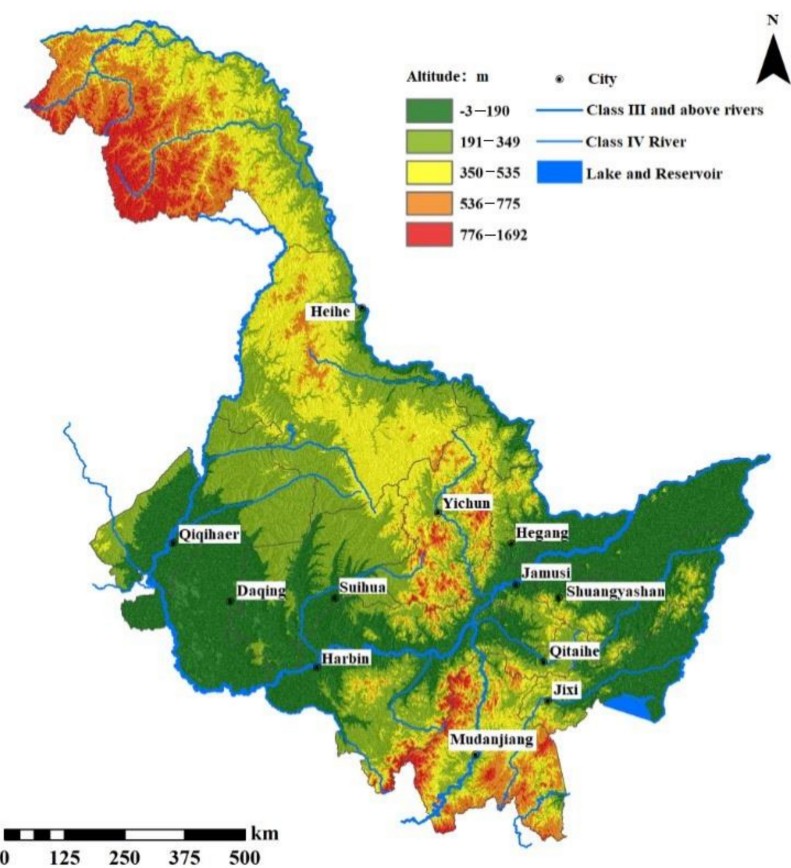

**Figure 1.** Profile of the study area.

### 2.2. Data Sources

The grain-sowing area, total agricultural machinery power, absolute amount of fertilizer, agricultural gross product, grain yield and other agricultural data were derived from the Heilongjiang Statistical Yearbooks published in 2008~2018 (the explanations of the indicators are presented in Table 1) [16,17]. The precipitation and other meteorological data were derived from the China Meteorological data network; the agricultural water consumption data, such as the gross irrigation water per unit area, were obtained from the statistical yearbook of water conservancy construction in Heilongjiang Province, and the grain irrigation quota was derived from the irrigation information network of Heilongjiang Province (http://www.hljggxx.gov.cn, last accessed: 15 March 2022). The above data are annual totals.

**Table 1.** Explanation of the indicators.

| Indicators | Explanation |
| --- | --- |
| Grain-sowing area/hm$^2$ | The area actually sown or transplanted with grains throughout the year |
| Total agricultural machinery power/KW | The sum of the rated power of all agricultural machinery |
| Absolute amount of fertilizer/Kg | The converted amount according to 100% nitrogen, phosphorus pentoxide and potassium oxide concentrations |
| Agricultural gross product/Yuan | The total value of all agricultural products expressed in currency and various supporting service activities for the agricultural production activities |

*2.3. Water Footprint Theory*

The water footprint theory involves the development of virtual water, which characterizes the amount of freshwater resources that is required to produce a product or provide a service in a certain period of time, including blue water, green water and gray water. Blue water is defined as the water contained in rivers, lakes and groundwater; green water is mainly precipitation; and gray water is the amount of freshwater needed to absorb pollutants [12,17].

2.3.1. Grain Water Footprint

Grains are water-intensive products, and the grain water footprint is usually calculated according to the actual amount of water consumed per grain yield unit in the grain-planting process [18]. In this research, the grain water footprint consists of the blue and green water footprints of grain. The blue water footprint of grain refers to the consumption of surface water and groundwater that can be used by grains, while the green water footprint of grain refers to the precipitation that can be used by grains [17]. The grain water footprint formula can be expressed as follows:

$$w_f = w_{fb} + w_{fg} \tag{1}$$

$$\begin{cases} w_{fb} = 10\dfrac{I_g}{y} \\ w_{fg} = 10\dfrac{P_e}{y} \end{cases} \tag{2}$$

where $w_f$ is the grain water footprint, in m$^3$/t; $w_{fb}$ is the grain blue water footprint, in m$^3$/t; $w_{fb}$ is the grain green water footprint, in m$^3$/t; $I_g$ is the gross irrigation water per unit area during the grain growth period, in mm; $y$ is the grain yield, in t/hm$^2$; and $P_e$ is the effective precipitation during the grain growth period. Its calculation is referenced from the Food and Agriculture Organization of the United Nations as follows, where $T$ is rainfall days [19]:

$$P_e = \begin{cases} \dfrac{PT(4.17 - 0.2P)}{4.17} & (P < 8.3\text{mm/d}) \\ T(4.17 + 0.1P) & (P \geq 8.3\text{mm/d}) \end{cases} \tag{3}$$

2.3.2. Grain Gray Water Footprint

The gray water footprint was proposed by Hoekstra to measure the freshwater resources required to dilute pollutants to a standard water quality [20]. In this research, the grain gray water footprint is defined as an unexpected output index, and nitrogen fertilizer is used as the main pollutant in the grain gray water footprint to measure the water pollution caused by nitrogen in the fertilizer that is input in the grain-planting. The formula used to calculate the grain gray footprint is as follows:

$$w_g = \frac{(\alpha \times AR)}{(C_{max} - C_{nat})} \tag{4}$$

where $w_g$ is the grain gray water footprint; $\alpha$ is the nitrogen fertilizer leaching rate, and by referring to relevant research, the leaching rates of rice, soybean and maize in Heilongjiang Province are found to be 14%, 5% and 12%, respectively; *AR* is the nitrogen application rate, the average values of which for rice, soybean and maize are 136, 113 and 225 kg/hm$^2$, respectively; $C_{max}$ is the maximum permissible concentration of nitrogen fertilizer, derived using the United States Environmental Protection Agency (EPA) standard of 0.01 kg/m$^3$; and $C_{nat}$ is the natural tolerance of nitrogen fertilizer, the value of which is 0 kg/m$^3$ [21,22].

*2.4. SBM-DEA Model*

DEA is a nonparametric technical efficiency analysis method that is based on relative comparisons between decision-making units (DMUs); this method was proposed by Charnes, Cooper and Rhodes in 1978, does not require dimensional unity, and has a relatively simple principle. It has advantages when applied to multi-input–output systems and is widely used in many fields [23,24]. The traditional DEA model has two shortcomings.

(1) The efficiency of this method has been compared and analyzed only from the effective frontier, leading to the problem in which DMUs are not sorted when they are all in the effective frontier. (2) This method cannot solve the problem of relaxation in an input–output system. However, the SBM–DEA model can solve these problems of radial selection and input–output relaxation [25].

Therefore, the SBM-DEA model was selected in this research to analyze the gaps and change trends associated with irrigation water use efficiency among 11 cities in Heilongjiang Province compared with the frontier: one city is regarded as one DMU. At this time, 11 DMUs exist in the system, and 5 inputs, 1 expected output and 1 unexpected output exist in each DMU, with *xij* (*I* = 1, 2..., 5; *j* = 1, 2..., 11) representing the input of the *j*-th DMU to the *i*-th product and *yrj* (*r* = 1, 2; *J* = 1, 2..., 11) representing the output of the *j*-th DMU to the *r*-th product. The model is constructed as follows:

$$min\rho = \frac{1 - \frac{1}{N}\sum_{i=1}^{N}\frac{s_i^-}{x_{io}}}{1 + \frac{1}{(M+I)}\left(\sum_{r=1}^{M}\frac{s_{r1}^+}{y_{ro}^1} + \sum_{r=1}^{I}\frac{s_{r2}^+}{y_{ro}^2}\right)}, \tag{5}$$

$$s.t. \begin{cases} \sum_{j=1}^{K}\lambda_j x_{ij} + s_i^- = x_{io} \\ \sum_{j=1}^{K}\lambda_j y_{ij}^1 - s_{r1}^+ = y_{ro}^1 \\ \sum_{j=1}^{K}\lambda_j y_{ij}^2 - s_{r2}^+ = y_{ro}^2 \\ \lambda_j \geq 0; s_i^- \geq 0; s_{r1}^+ \geq 0; s_{r2}^+ \geq 0 \end{cases} \tag{6}$$

where $\rho$ is the efficiency, *N* is the total number of input variables, *M* is the total number of expected outputs, *I* is the total number of unexpected outputs, $x_{io}$ and $y_{ro}$ are the input and output variables of the DMUs, respectively, $s_i^-$ and $s_r^+$ are the input and output relaxation variables, respectively, $\lambda_j$ is the weight, and *K* is the total number of DMUs.

When $\rho = 1, s_i^- = 0$, and $s_r^+ = 0$, the DMU is effective and has a high water use efficiency; when $0 < \rho < 1$, the efficiency is not high and can be improved by optimizing $x_{io}$ and $y_{ro}$.

In the term $\rho$ = [MIX] $\times$ [TE], the mixing efficiency (MIX) is defined as the ratio of the SBM-derived technical efficiency (SMB TE) to the Chames, Cooper and Rhodes (CCR) technical efficiency (CCR TE). The technical efficiency is composed of the product of the pure technical efficiency (PTE) and scale efficiency (SE), as determined by the CCR model. The PTE and SE reflect a comparison among different DMUs in a certain period innovation

and management competitiveness in terms of the pure technology and scale technology, respectively, that transform the production factors into outputs [26].

## 3. Results

### 3.1. Analysis of the Grain Water Footprint in Heilongjiang Province

According to the grain water footprint definition, the water footprints and gray water footprints of rice, soybean and maize in 11 cities in Heilongjiang Province were calculated over the growth period from 2008 to 2018.

Temporal Distribution

Figure 2 shows that the grain water footprint of Heilongjiang Province showed a downward trend from 2008 to 2018; a slow upward trend was identified from 2008 to 2010, with an increase of 20.2%. After a sudden drop to 1708.56 $m^3/t$ in 2011, the total grain water footprint remained at approximately 2000 $m^3/t$ from 2012 to 2017 until it recovered with an obvious upward trend in 2018. The changes in the total grain water footprint were related to the grain yield, natural climate conditions and planting structure. As determined through further comparisons and analyses of the blue and green water footprints of grain, their trends were consistent with that of the total grain water footprint, and the ratio of the blue water footprint of grain to the green water footprint of grain always tends toward 1:3, indicating that the agricultural water structure of Heilongjiang Province has remained stable over the last 11 years.

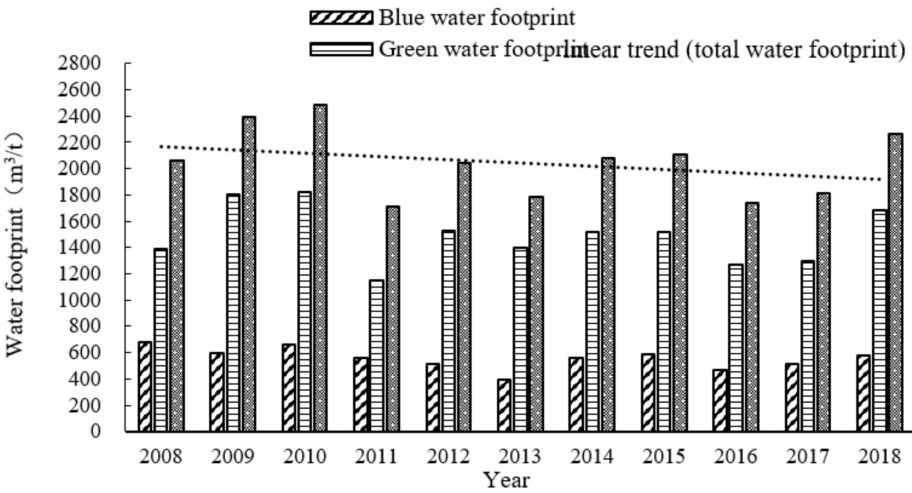

**Figure 2.** Temporal distribution of grain green, blue and total water footprint in Heilongjiang Province in 2008–2018.

In Figure 3, the total grain gray water footprint of Heilongjiang Province showed a slow upward trend followed by a downward trend from 2008 to 2017, ranging from the lowest point of $1.209 \times 10^9$ $m^3$ in 2008 to the highest point of $1.995 \times 10^9$ $m^3$ in 2015, an increase of 64.9%; then, the grain gray water footprint decreased for two years (2016 and 2017) to $1.610 \times 10^9$ $m^3$ before slightly recovering to $1.773 \times 10^9$ $m^3$ in 2018.

As determined from the grain structure comparison, the grain gray water footprint in Heilongjiang Province is most affected by corn planting and least affected by soybean planting. This is because nitrogen leaching rates differ among different grains, and the nitrogen leaching rate of soybean is only 5%, so the water pollution resulting from soybean agriculture is far less than that resulting from the cultivation of other grains.

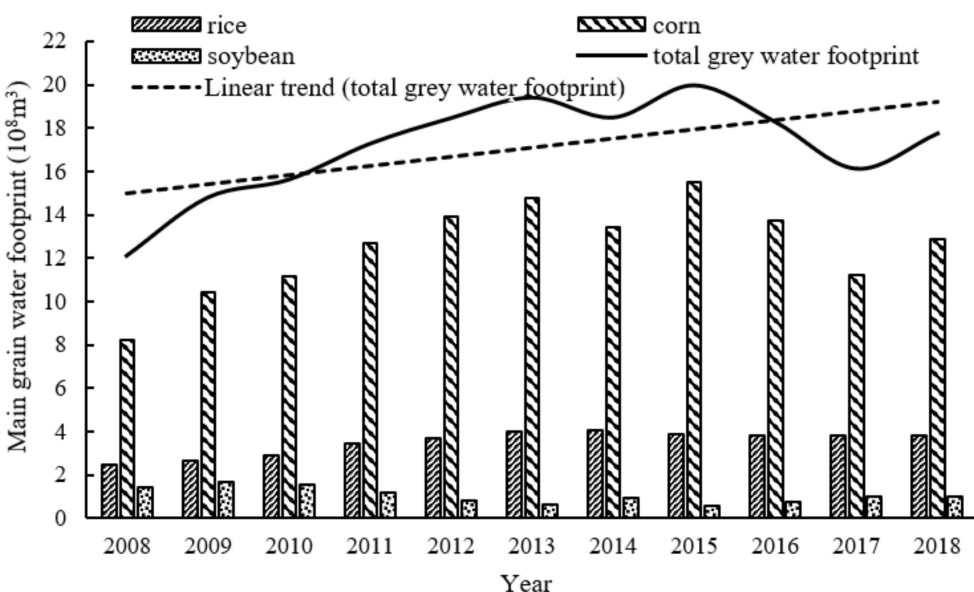

**Figure 3.** Temporal distribution of grain gray water footprint in Heilongjiang Province in 2008–2018.

Figure 4 shows that the total grain water footprint varies greatly from region to region, with an uneven spatial distribution and situations in which grain water footprints characterizing excess and scarcity coexist. The total grain water footprint of the whole province generally shows a low trend in the southeast and a high trend in the northwest. Low-grain-water-footprint areas are mainly concentrated in the eastern and southern regions, including in Jixi, Jiamusi, Daqing, Suihua and Harbin, among which Harbin has the lowest grain water footprint. High-grain-water-footprint areas are mainly concentrated in the northern, western and southern regions, including Yichun, Mudanjiang, Qitaihe and Qiqihar, among which Yichun has the highest total grain water footprint.

In Figure 5, the gray water footprint of grain in the whole province shows a trend of being low in the northeast and high in the southwest. Low-value areas are mainly concentrated in the northern and eastern regions, including Yichun, Hegang, Shuangyashan and Qitaihe, among which Yichun has the gray water footprint of grain. High-value areas are mainly concentrated in the southern and western regions, including Qiqihar, Harbin and Suihua, among which Harbin has the highest gray water footprint of grain. The extreme values of the gray water footprint of grain show the opposite trends of those identified for the blue and green water footprints of grain.

*3.2. Evaluation of Irrigation Water Use Efficiency in Heilongjiang Province*

The collected indicators (the grain sowing area, total power of agricultural machinery, grain blue water footprint, grain green water footprint, net amount of chemical fertilizer, total output value of agriculture, forestry, animal husbandry and fishery, and grain gray water footprint) were input into the SBM-DEA model using DEA-SOLVER pro 5.0 software; then, the irrigation water use efficiencies of 11 cities in Heilongjiang Province were calculated (Table 2). It can be seen from Table 1 that the annual average irrigation water efficiency of Harbin, Qiqihar, Jixi, Suihua, Daqing and Qitai River are higher than the provincial average, while Jiamusi, Hegang, Shuangyashan, Yichun and Mudanjiang are lower than the provincial average, and Mudanjiang has the lowest annual average irrigation water use efficiency.

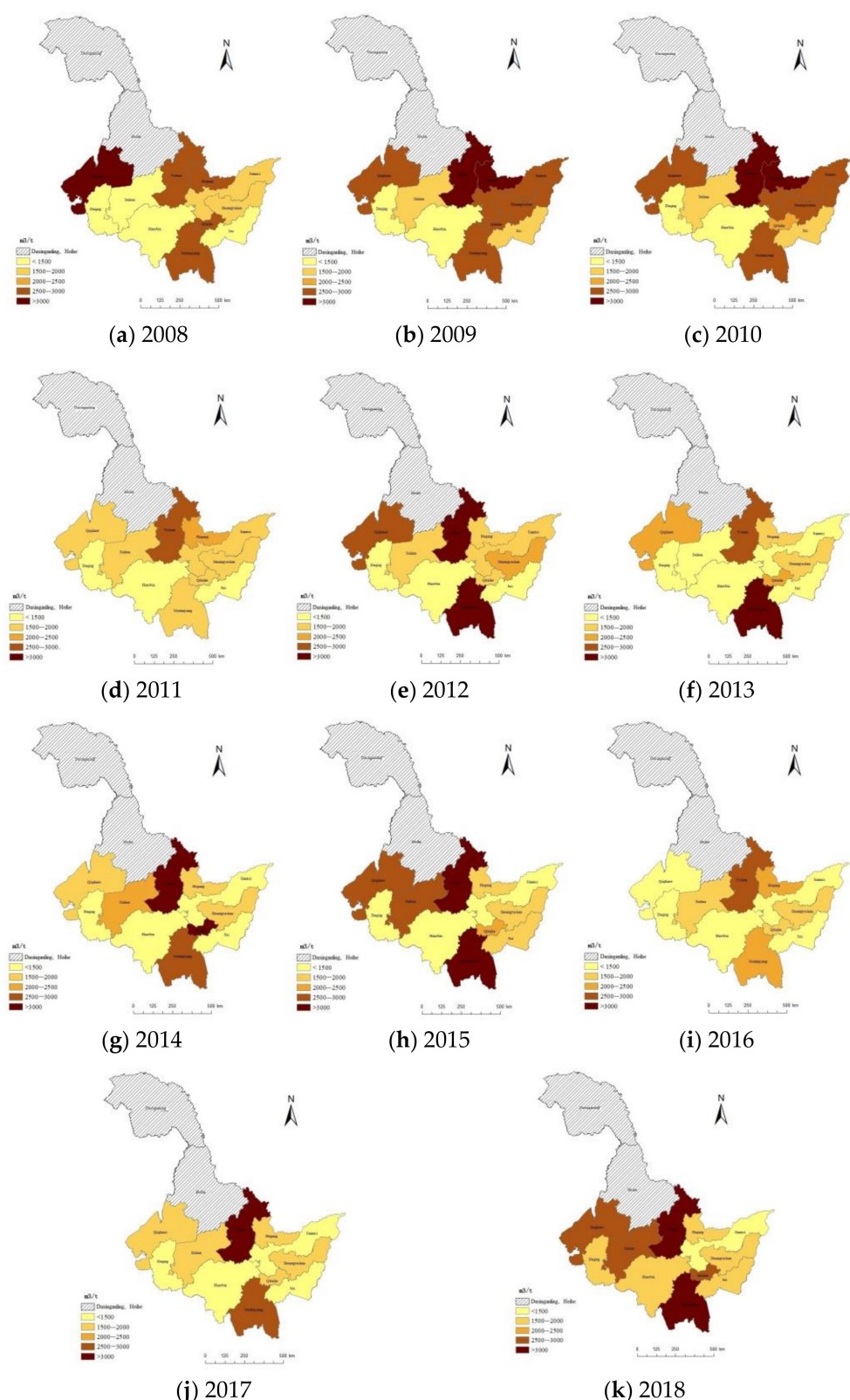

**Figure 4.** Annual spatial distribution of the total grain water footprint in Heilongjiang Province.

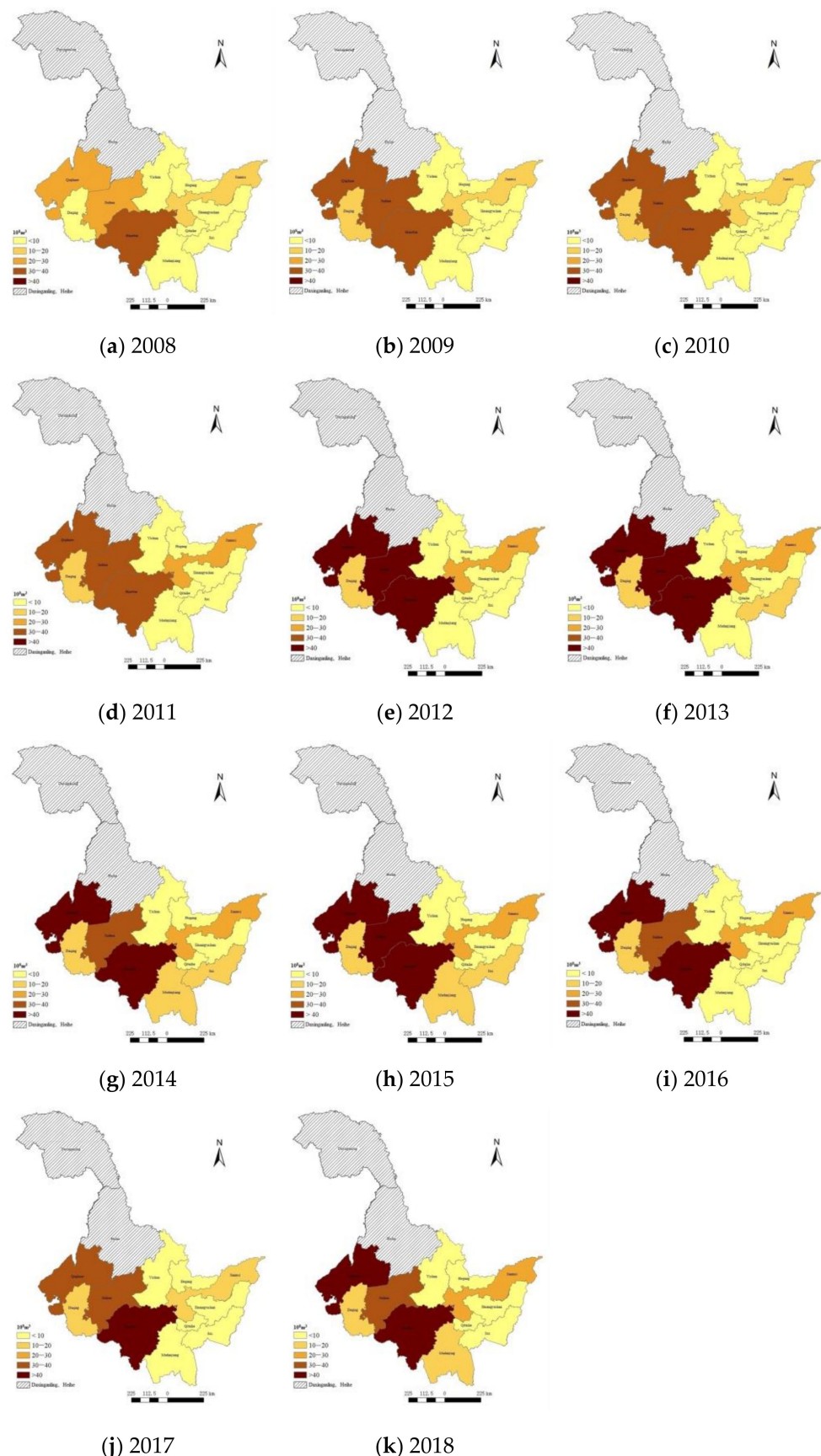

**Figure 5.** Annual spatial distribution of grain grey water footprint in Heilongjiang Province.

**Table 2.** Statistical results of irrigation water efficiency of 11 cities in Heilongjiang Province.

| City | Annual Average Irrigation Water Efficiency | Gap from the Provincial Average/% |
|---|---|---|
| Harbin | 1.000 | 21.73% |
| Qiqihaer | 1.000 | 21.73% |
| Jixi | 1.000 | 21.73% |
| Suihua | 0.937 | 14.10% |
| Daqing | 0.883 | 7.52% |
| Qitaihe | 0.857 | 4.27% |
| Jiamusi | 0.744 | −9.38% |
| Hegang | 0.716 | −12.85% |
| Shuangyashan | 0.691 | −15.89% |
| Yichun | 0.674 | −18.00% |
| Mudanjiang | 0.534 | −34.96% |

To facilitate the integration of regional analyses, Heilongjiang Province was divided into four regions according to the eastern, western, southern and northern directions, and the agricultural water use efficiency values of each region in Heilongjiang Province from 2008 to 2018 were obtained, as shown in Figure 6. (1) A large gap in regional irrigation water use efficiency was identified from 2008 to 2015, and the southern and western regions took the lead. The irrigation water use efficiency was low in the northern and eastern regions because Harbin and Qiqihar, as the forefront cities regarding irrigation water use efficiency in the studied province, drive the technical level of grain production in the southern and western regions. The Suihua River network has a high water density, and Daqing is rich in surface water resources, causing the water conservancy conditions in the southern and western regions to be better than those in other areas. (2) From 2016 to 2018, the gap in regional irrigation water use efficiency shrank, indicating that the water resource management capacity of the studied region was significantly improved, driving the regional irrigation water use efficiency gap to gradually narrow.

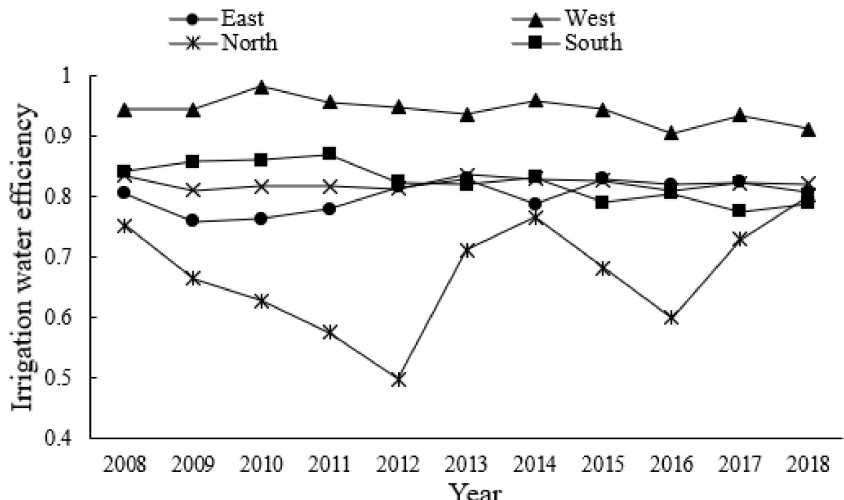

**Figure 6.** Irrigation water efficiency in Heilongjiang Province in 2008–2018.

In terms of time, the overall irrigation water use efficiency of Heilongjiang Province showed a steady and slow downward trend from 2008 to 2018. The comprehensive efficiency was not high, and the average irrigation water use efficiency value was between 0.7 and 1.0; the relative efficiency was high in 2014. In terms of space, the irrigation water use efficiencies of Harbin, Qiqihar and Jixi were the best, while the irrigation water use efficiencies of Suihua, Daqing and Qitaihe were above the moderate level. These results show that Harbin, Qiqihar and Jixi are at the forefront of irrigation water use efficiency among the whole province. With economic and social development and the improvement

of agricultural productivity, the gap between the average grain production technology level in Heilongjiang Province and the frontier cities is gradually widening, so the irrigation water use efficiency has decreased relative to the frontier level.

### 3.3. Spatial Distribution

ArcGIS was used to draw a map of the spatial distribution of the irrigation water use efficiencies of 11 cities in Heilongjiang Province from 2008 to 2018, as shown in Figure 7. If the irrigation water use efficiency is 1, the water use efficiency has reached the effective state; if it is less than 1, the irrigation water use efficiency is not effective [27]. The average irrigation water use efficiency in Heilongjiang Province is 0.821, indicating that the irrigation water use efficiency is not effective in the province as a whole. The spatial distribution of regional irrigation water use efficiencies is uneven, and the average irrigation water efficiencies in the eastern, western, southern and northern regions were 0.802, 0.942, 0.824 and 0.674, respectively, showing the pattern of west > south > east > north.

The irrigation water use efficiency is divided into five grades: the irrigation water use efficiencies of Harbin, Qiqihar and Jixi are level-I, indicating that their agricultural water resources have been fully utilized and that optimal allocation has been realized. The irrigation water use efficiencies of Daqing, Suihua and Qitaihe are level-II, which means that full use has been made of the agricultural water resources in these regions, but there is still room for improvement. The irrigation water use efficiencies of Jiamusi and Hegang are level-III, revealing a certain gap with the provincial average level and that these regions still have great water-saving potential. The irrigation water use efficiencies of Shuangyashan and Yichun are level-IV, suggesting that the irrigation water use efficiencies in these regions need to be greatly improved. Finally, the irrigation water use efficiency of Mudanjiang is grade-V, far lower than the average level obtained for the whole province.

### 3.4. Analysis of Underdeveloped Regions

Taking the average irrigation water use efficiency as the boundary condition, the average irrigation water use efficiencies of 11 cities in Heilongjiang province from 2008 to 2018 were divided into two levels; the first level includes Harbin, Qiqihar, Jixi, Daqing, Suihua and Qitaihe, and the average irrigation water use efficiencies in these regions are higher than the whole-province average. The second level comprises Jiamusi, Hegang, Shuangyashan, Yichun and Mudanjiang, the average irrigation water use efficiencies of which are lower than the whole-province average.

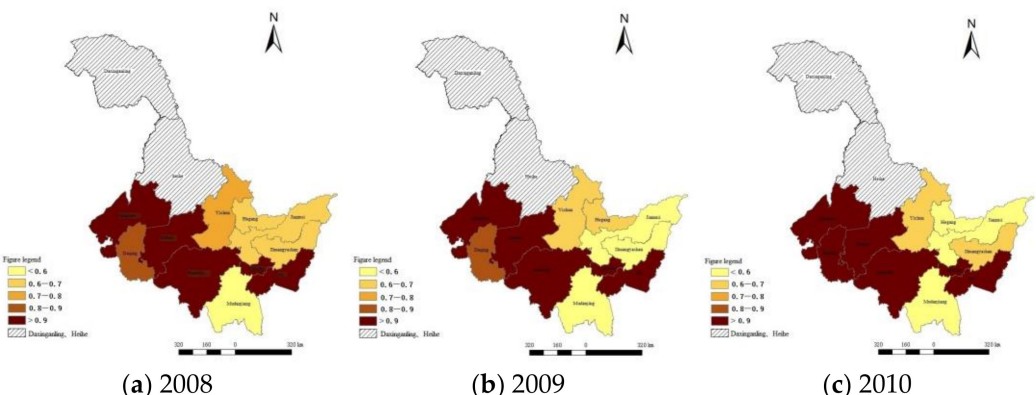

(**a**) 2008      (**b**) 2009      (**c**) 2010

**Figure 7.** *Cont.*

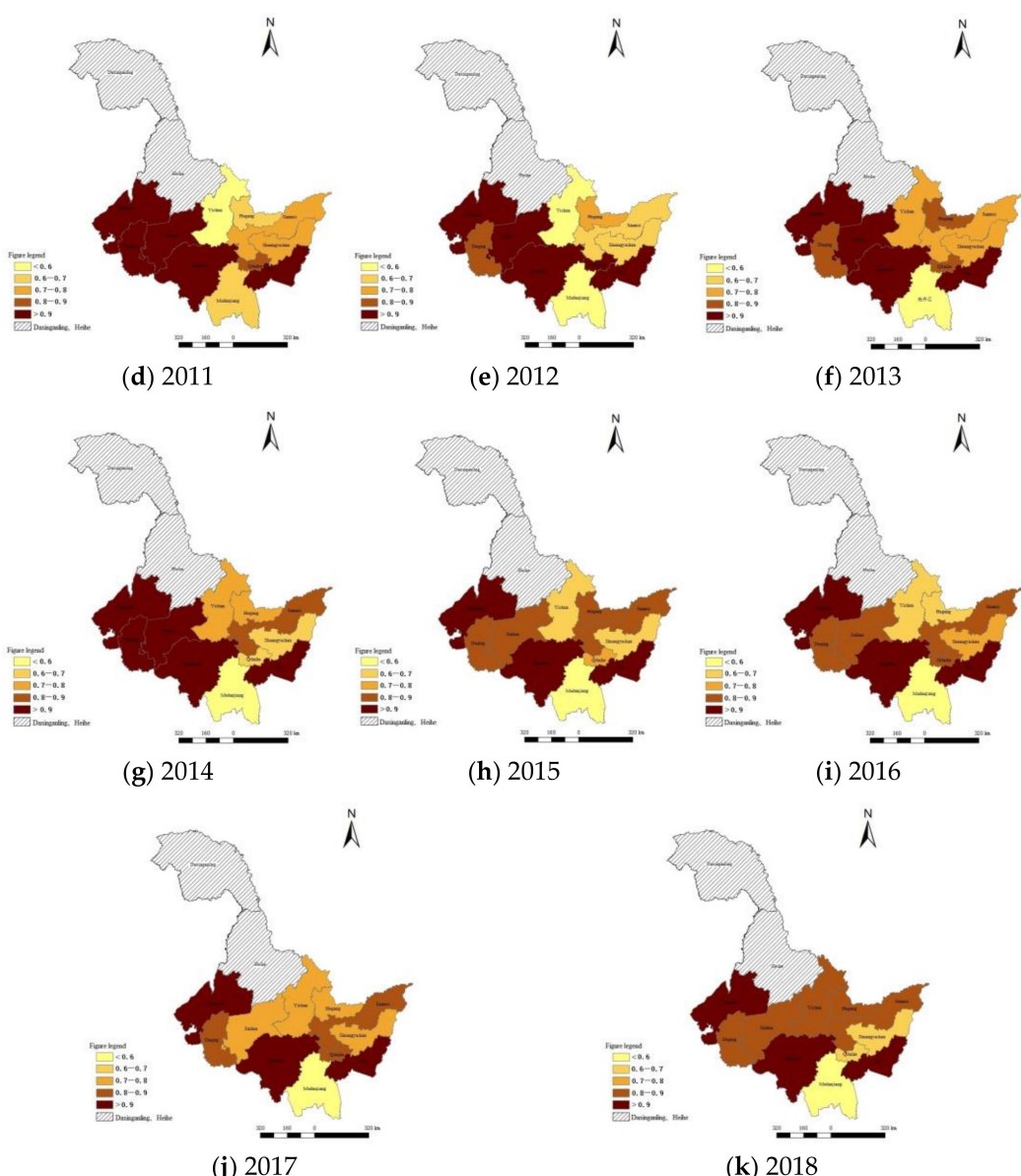

**Figure 7.** Annual spatial distribution of irrigation water efficiency in Heilongjiang Province.

We decomposed the irrigation water use efficiencies of the five second-level cities that do not reach the level of the provincial average using DEAP software to determine the reasons for their underdeveloped irrigation water use efficiencies (Table 3). According to relevant definitions, high pure technical efficiencies indicate high levels of agricultural production technology and water resource management, and high scale efficiencies indicate a high consistency of the input–output ratio in the system [28,29]. As seen from Table 3, the pure technical efficiency is greater than the scale efficiency in Jiamusi, but none of the values reached the provincial average level; the scale efficiency is increasing annually, and the irrigation water use efficiency mainly depends on the pure technical efficiency. The pure technical efficiency of Jiamusi needs to be strengthened, and the agricultural production technology should be improved. The pure technical efficiency of Hegang is greater than the scale efficiency and higher than the provincial average level, and the scale efficiency is far lower than the provincial average level; thus, we should focus on the large-scale production level of Hegang. The pure technical efficiency and scale efficiency of Shuangyashan are close to the average level across the whole province, and the scale efficiency basically maintains an annually increasing trend; we can learn from the advanced

technologies of developed areas, as the pure technical efficiency and scale efficiency of Yichun are close to 1, exceeding the provincial average levels, and the pure technical efficiency and scale efficiency Mudanjiang were high from 2008 to 2017, during which time the scale efficiency remained unchanged. In 2018, the scale efficiency decreased, and the pure technical efficiency was less than the scale efficiency. We should focus on improving the pure technical efficiency from the aspects of infrastructure construction, the mechanical level, the technical level and so on [30–33].

**Table 3.** Results of the decomposition of irrigation water efficiency in the 5 cities of the second level in 2008~2018.

| City | SBM TE | CCR TE | PTE | SE |
|---|---|---|---|---|
| Jiamusi | 0.744 | 0.830 | 0.932 | 0.890 |
| Hegang | 0.716 | 0.801 | 0.997 | 0.803 |
| Shuangyashan | 0.691 | 0.940 | 0.993 | 0.946 |
| Yichun | 0.674 | 1.000 | 1.000 | 1.000 |
| Mudanjiang | 0.534 | 0.991 | 0.992 | 0.998 |
| Provincial average level | 0.821 | 0.958 | 0.992 | 0.965 |

## 4. Discussion

### 4.1. Suggestions on Sustainable Development of Grain Water in Heilongjiang Province

In terms of time, the total grain water footprint of Heilongjiang Province showed a downward trend on whole in 2008–2018 (Figure 2), reached the lowest in 2017 and gradually increased in 2018. The causes of the trend were analyzed, including the following aspects: (1) industrial and domestic water occupied agricultural water, which promoted the improvement of agricultural modernization and the irrigation water efficiency; (2) global warming leads to the acceleration of evaporation and the reduction in precipitation which can be used in the process of grain production; (3) in 2018, the precipitation in Heilongjiang province increased significantly, resulting in the growth of grain blue water footprint and green water footprint, and the total water footprint increased. From 2008 to 2017, the grain grey water footprint of Heilongjiang Province showed a trend of slow rise and then decline which is due to the zero-growth policy of chemical fertilizer in Heilongjiang Province. The Chinese ministry of agriculture focused on the implementation of the zero-growth action of chemical fertilizer in 2015 [34]. Heilongjiang Province is strengthening the supervision of excessive fertilization in order to further improve its agricultural sustainable development ability; some effective measures have been taken to reduce the application of chemical fertilizer.

Spatially, the grain water footprint is low in the southeast and high in the northwest of Heilongjiang Province. The grain water footprint is rich in Yichun, Mudanjiang, Qitaihe and Qiqihar, while it is relatively low in Jixi, Jiamusi, Daqing, Suihua and Harbin; the grain grey water footprint shows a trend of low in the northeast and high in the southwest. For the grain water footprint, engineering measures should be taken to realize the "water transfer from northwest to southeast" in Heilongjiang Province, and the water resources in the areas with excess water should be transferred to the areas with scarce water, so as to make spatial distribution of grain water footprint more uniform in Heilongjiang Province. For the grain grey water footprint, the main reason for the trend of the grain gray footprint is the large amount of maize and soybean planting in the west, and the dependence of the corn on chemical fertilizer is higher; the southwest of Heilongjiang Province should reduce its dependence on chemical fertilizer, continue to strengthen the supervision of excessive fertilization and optimize the application of chemical fertilizer [10].

### 4.2. Relationship between Irrigation Water Use Efficiency and Socio-Economic Factors

The gap of irrigation water use efficiency among 11 cities gradually narrowed during 2008–2018. The irrigation water management of Heilongjiang Province has been steadily improved. The average of irrigation water use efficiency is between 0.7 and 1.0, of which the relative efficiency in 2014 is highest. The distribution of irrigation water use efficiency

in Heilongjiang Province is also uneven, showing a pattern of west > south > east > north. The average of irrigation water use efficiency in the east, west, south and north is 0.802, 0.942, 0.824 and 0.674, respectively.

The reasons for the significant spatial differences identified in the irrigation water use efficiency among regions in Heilongjiang Province are described as follows. (1) Qiqihar, located in the western part of Heilongjiang Province, and Harbin, Daqing, and Suihua, located in the southern part, are suitable for large-scale mechanized planting due to the influence of the local topographic and climate conditions [31]; these regions are located in arid and semiarid areas where the precipitation conditions are poor [35], and the limited water resources are used effectively, causing the irrigation water use efficiencies to be relatively high. (2) Qitaihe, Jiamusi, Hegang and Shuangyashan, located in eastern Heilongjiang Province, are rich in precipitation, flat terrain and fertile lands; these areas are suitable for large-scale mechanized planting and distributed farmland irrigation facilities and have relatively developed water-saving irrigation technologies, allowing their irrigation water use efficiencies to be relatively high [36]. (3) In Jixi in the eastern region and Mudanjiang in southern region, the landforms are mainly hilly and mountainous, with rich precipitation resources and optimal irrigation measures. The reason the irrigation water use efficiencies of these regions are separated into high and low value areas in the province involves the grain yield differences that result from the grain cultivation structures; the difference in the grain water footprints between the two regions is obvious. The multiyear average grain water footprint per unit of grain output in Jixi is 1510.94 $m^3$/t, while that in Mudanjiang is 2699.01 $m^3$/t.

All cities should reform considering their own lack of efficiency, rather than develop irrigation water management according to an established model. The main reason for the low irrigation water use efficiency in Hegang is the lack of scale efficiency. At present, the urgent task for Hegang is to expand the agricultural production scale and promote the modern agriculture; for Jiamusi with low pure technical efficiency and scale efficiency, it should reform with the improvement of management level and the expansion of agricultural production scale; Mudanjiang's pure technical efficiency and scale efficiency were high from 2008 to 2017, the scale efficiency decreased in 2018, and the pure technical efficiency was lower than the scale efficiency. It is suggested to pay attention to the improvement of pure technical efficiency from the aspects of infrastructure construction, agricultural machinery and technical level.

### 4.3. Future Research

Due to the limitation of data, nitrogen fertilizer was mainly used for calculating gray-water footprint in this research; therefore, there may be some deviations. Future research will combine other indicators to comprehensively characterize the gray water footprint and establish a comprehensive indicator system that can systematically characterize the grain gray water footprint.

In this research, the effects of factors of the grain sowing area, total power of agricultural machinery, grain blue water footprint, grain green water footprint, net amount of chemical fertilizer, total output value of agriculture, forestry, animal husbandry and fishery, and grain gray water footprint on irrigation water use efficiency were considered. However, there are many factors affecting irrigation water use efficiency in actual production, such as planting structure, crop management measures, etc. With the deepening of investigation and the gradual enrichment of data, the research on the influencing factors of irrigation water use efficiency will be further expanded.

## 5. Conclusions

From the perspective of the water footprint, the input–output data of 11 municipalities in Heilongjiang Province from 2008 to 2018 were used to calculate the irrigation water use efficiency of Heilongjiang Province through the SBM-DEA model; based on the results, the spatial and temporal distribution patterns and influencing factors were analyzed, and the

technical efficiencies of the five cities with second-level irrigation water use efficiencies were decomposed. The main conclusions are described below.

(1)  In view of the timeline, the irrigation water use efficiency of Heilongjiang Province showed a steady and slow downward trend from 2008 to 2018 with an average irrigation water efficiency of 0.821; this level did not reach the effective state. Large gaps were found in irrigation water use efficiency among different regions from 2008 to 2015, and the gap in the regional irrigation water use efficiency gradually narrowed from 2016 to 2018.

(2)  In view of the spatial distribution, significant differences were found in the irrigation water use efficiencies in different regions of Heilongjiang Province, showing the overall distribution pattern of west > south > east > north. Harbin, Qiqihar and Jixi had high-level irrigation water use efficiencies among the whole province; Jiamusi, Hegang, Shuangyashan, Yichun and Mudanjiang had second-level irrigation water use efficiencies that did not reach the provincial-level average.

(3)  At the second level, the pure technical efficiency was greater than the scale efficiency in Jiamusi and Hegang. The pure technical efficiency and scale efficiency in Shuangyashan were both close to the corresponding average levels of the whole province. The pure technical efficiency and scale efficiency were close to 1 in Yichun. In Mudanjiang, the pure technical efficiency was less than the scale efficiency only in 2018. The irrigation water use efficiency mainly depends on the pure technical efficiency. We should learn from the regions with advanced technology and gradually improve the technical level throughout the province.

(4)  To address the problems associated with agricultural water use in Heilongjiang Province, on the one hand, we should strengthen interregional cooperation and government macrocontrol, optimize the agricultural production planting structure according to regional characteristics, and improve the water and land resource allocation efficiencies. Additionally, we should actively develop agricultural science and technology research, optimize the agricultural water use mode, and increase investment in agricultural technologies and water conservancy construction in irrigated areas to improve the irrigation water use efficiency by enhancing the pure technical efficiency.

**Author Contributions:** Conceptualization and methodology, W.L.; software, data curation, Z.M.; visualization, B.L. All authors have read and agreed to the published version of the manuscript.

**Funding:** This research was funded by National Natural Science Foundation of China (52109049); Natural Science Foundation of Heilongjiang Province (LH2019E009).

**Institutional Review Board Statement:** Not applicable.

**Informed Consent Statement:** Not applicable.

**Data Availability Statement:** Not applicable.

**Conflicts of Interest:** The authors declare no conflict of interest.

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
