# Peer review of "Spatiotemporal Distribution of Irrigation Water Use Efficiency from the Perspective of Water Footprints in Heilongjiang Province"

_water, doi:10.3390/w14081232_

Round 1

Reviewer 1 Report

GENERAL COMMENTS TO AUTHORS:

In general, the paper is well-structured, but some details should be explained in the document.

In the attached document authors will be able to see the comments from the Reviewer. Special clarifications should be done: 

  1. It is necessary to clarify adequately in the Introduction a contextualization of the crop area and the irrigation technology used.
  2. As well, some of the asseverations done in the document (Results and discussion) are too general to justify the water footprint trends. Please, give a more specific discussion.

Reviewer 2 Report

The present article is a very interesting one. The abstract is very well developed and includes relevant information on the main content.

The Introduction section need to be better sustained with international references. Currently, most of these references are only from China. Please analyze the problem that you approached from an international perspective.

Section 2 - Materials and Methods

A graph/ figure based on the used methodology will improve the quality of this section.

Results

This section can be improved by increasing the quality of figures 4, 5 and 7 which are hardly readable. 

Subsection 3.2 can be transferred to next section (4 - Discussions) which is underdeveloped in the present form.

The results presented in table 1 should be better explained.

Section 4 should be improved by transfering parts from section 3.

Overall, the article is well structured/ developed and only some minor corrections are need it.
